# PCA-YOLO: A Small Liver Tumor Detection Model with Patch-Contrastive Attention

**Xueyang Li**[1]                                                    XLI34@ND.EDU

[1] *University of Notre Dame, Notre Dame, USA*

**Han Xiao**[2]                                                    XIAOH69@MAIL.SYSU.EDU.CN

[2] *The First Affiliated Hospital of Sun Yat-sen University, Guangzhou, China*

**Zongpeng Weng**[2]                                         WENGZP5@MAIL2.SYSU.EDU.CN

**Xinrong Hu**[1]                                                   XHU7@ND.EDU

**Danny Chen**[1]                                                DCHEN@ND.EDU

**Yiyu Shi**[1]                                                      YSHI4@ND.EDU

**Editors:** Accepted for publication at MIDL 2025

## Abstract

Liver tumors, as one of the most common malignant tumor types, represent a significant clinical challenge, with the detection of small tumors being particularly problematic. Despite the rapid advances in deep learning (DL) offering significant support in reducing the workload of radiologists, current detection models still struggle with the detection of small tumors. This is particularly troubling as these are the cases where even experienced radiologists are more prone to errors, underscoring the critical need for improved accuracy of detection methods in this area. Addressing this critical gap, this article introduces patch-contrastive attention YOLO (PCA-YOLO), an innovative adaptation of the YOLO framework, incorporating a patch-based attention module to specifically target the detection of small liver tumors. Furthermore, we collected a specialized CT dataset focusing exclusively on small liver tumors, complemented with meticulously annotated bounding boxes, to facilitate this study. Our experimental findings demonstrate that our approach achieves a leading mean Average Precision (mAP) score of 77.2% at a 50% Intersection Over Union (IoU) threshold, surpassing all current leading detection methods tested against our specialized dataset.

**Keywords:** Liver tumor detection, Attention mechanism, CT scan dataset, YOLO model.

## 1. Introduction

Liver cancer ranks as the second most deadly form of tumor, with Hepatocellular Carcinoma (HCC) being its most prevalent type, constituting approximately 85% of all liver cancer cases (Villanueva, 2019). Computed tomography (CT) scanning is a fundamental tool for diagnosing HCC, but the manual evaluation of these scans is notably time-consuming. This is especially true for small liver tumors, on which the task of interpretation demands radiologists to have substantial experience and specialized knowledge.

On the other hand, advances in artificial intelligence have provided deep learning (DL) based approaches for liver tumor analysis using CT scans. Based on the widely recognized BCLC guideline and RECIST standard, the number of tumors is a critical parameter for staging HCC, and the maximum tumor diameter is a pivotal factor influencing patient prognosis (Eisenhauer et al., 2009; Reig et al., 2022). Given their capacity to efficiently derive

such critical information and the significantly reduced human effort needed for labeling, detection models stand out as more suitable alternatives for DL-aided HCC diagnosis, in comparison to segmentation models.

While liver tumor segmentation has garnered significant attention, the task of liver tumor detection remains relatively under-explored. Current approaches predominantly adapt modified U-Net (Ronneberger et al., 2015) structures (e.g., RA-Net (Kalsoom et al., 2022) and CLIP (Liu et al., 2023)) for this purpose. In addition to this, state-of-the-art detection models originally designed for natural images, like DETR (Carion et al., 2020) and YOLO (Redmon et al., 2016), show promise for medical imaging applications following certain adaptations, as evidenced by RCS-YOLO in brain tumor detection (Kang et al., 2023) and SPN-TS in breast tumor detection (He et al., 2023). However, a notable barrier to adapting these natural image detection models for medical settings is their performance with small objects. In natural scenes, objects typically occupy larger portions of the images and have more distinct features compared to tumors in medical scans. This issue becomes particularly acute when identifying small-sized tumors, demanding the model to further detect subtle, less pronounced features amidst the complex background of CT slices (Abdusalomov et al., 2023; He et al., 2023). Further, accurate detection of such small tumors is crucial, surpassing the importance of identifying larger, more noticeable tumors. This is because even experienced radiologists are more susceptible to mistakes when evaluating small tumors, and they may not be able to easily rectify the inaccuracies incurred by DL models, as they might with more conspicuous tumors.

Inspired by the successes of attention mechanism (Vaswani et al., 2017) and the contrastive learning strategy in the Siamese network (Koch et al., 2015), this paper synthesizes these methodologies to introduce a novel YOLO framework called *patch-contrastive attention YOLO* (PCA-YOLO), aiming to address the challenge of detecting small tumors. The rationale for this design is clear: While small tumors may be less conspicuous due to limited information, they should still be more distinguishable and tumor-like in appearance compared to other regions within the same slice. Additionally, we introduce to the public via this study our Small Liver Tumor Detection (SLTD) CT dataset, a pioneering resource specifically designed for detecting small liver tumors, to further promote research along this direction. Our PCA-YOLO approach attains an mAP@50% score of 77.2% on this dataset, surpassing the performance of all existing state-of-the-art detection models.

## 2. The SLTD Dataset

The liver tumor segmentation benchmark (LiTS) dataset (Bilic et al., 2023), primarily designed for segmentation, is commonly used when performing liver tumor detection (Liu et al., 2023). But, it poses challenges in converting segmentation masks to bounding boxes, especially in complex scenarios involving tumor clusters or adjacent tumors, where domain expertise is crucial. This is pivotal as tumor count directly influences HCC staging (Reig et al., 2022). To address this gap and provide a more dedicated and realistic dataset for the small liver tumor detection task, we introduce our Small Liver Tumor Detection (SLTD) dataset. The SLTD dataset comprises 208 3D CT volumes, totaling 41,587 2D slices. The CT scans are acquired during the portal venous phase, as it is more commonly used and clinically effective for tumor diagnosis compared to the arterial and unenhanced plain scan

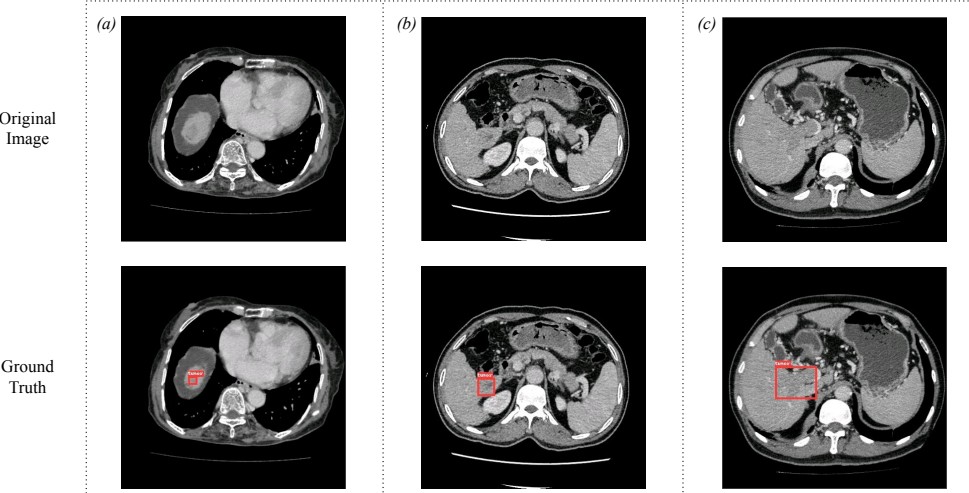

Figure 1: Representative CT slices from the SLTD dataset: (a) A tumor with a bounding box size of 0.8 cm; (b) a tumor with a bounding box size of 2.2 cm; (c) the largest tumor with a bounding box size of 5.0 cm.

phases. The images have a resolution of 512×512 and were acquired from the same CT machine, with a window width of 150, window level of 50, radiation dose 120kV, slice thickness of 1 mm, and slice gap of 0.8 mm. The images underwent manual quality control to exclude any scans with noticeable artifacts or blurriness and to verify the completeness of all slices. From this collection, our team of radiologists with at least five years of clinical expertise meticulously selected and annotated 452 2D slices. These slices were chosen based on their proximity to the initial and final slices where a tumor is visible within each 3D volume. To ensure annotation reliability, only slices with bounding boxes size of at least 0.4 cm were included. In summary, our dataset contains bounding boxes with sizes ranging from 0.4 cm to 5.0 cm, while the tumor maximum diameters range from 1.0 cm to 6.5 cm. Each slice contains between 1 to 4 tumors, averaging 1.49 tumors per slice, providing a diverse and comprehensive representation of small tumor cases. Representative images from the selected slices are shown in Fig. 1. The dataset is publicly available at https://github.com/XLIAaron/Small_LiverTumor.

## 3. Methodology

The overall architecture of PCA-YOLO is depicted in Fig. 2(a), featuring a patch-contrastive attention (PCA) module at its core. Within this module, a Siamese network (Koch et al., 2015) is utilized to assess whether a given patch is similar to the original image, i.e., whether the patch contains a tumor. Following the PCA module, the architecture is integrated with a detection head, employing the YOLOv8x (Jocher et al., 2023) model for this purpose. The details are given below.

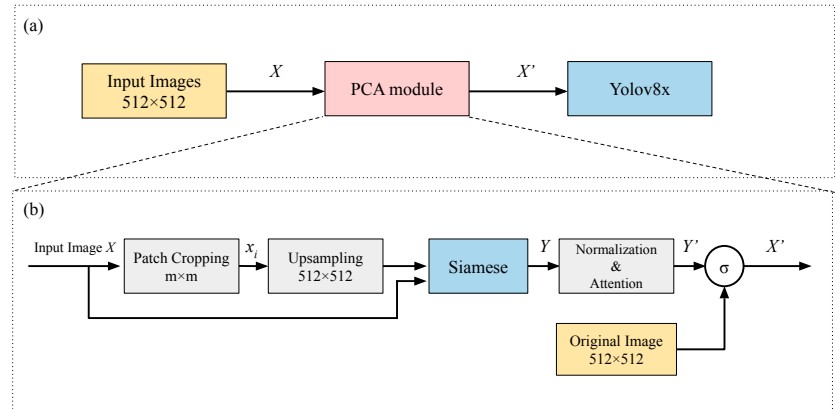

Figure 2: (a) The general structure of PCA-YOLO. $X$ is an original input image, and $X'$ denotes the attention-augmented input. (b) The detailed structure of the attention module. $x_i$ denotes the patches extracted using non-overlapping cropping of the original image $X$. The Siamese network is trained using each up-sampled patch and $X$ as a paired input, and the ground truth similarity is 1 if a patch contains any tumor and 0 otherwise. $Y$ is the collection of outputs from the Siamese network for all the patches of $X$, which is used to generate the attention map $Y'$. $Y'$ is concatenated with $X$ to form an attention-augmented input $X'$ for YOLOv8x.

### 3.1. Patch-Contrastive Attention Module:

Fig. 2(b) illustrates the design of the PCA module. The process begins with an input image $X$ of dimensions $512 \times 512$, which is cropped into $n$ smaller patches using a sliding window technique with no overlap, each patch $x_i$ being $m \times m$ in size, in which $m$ is a hyperparameter and its impact will be studied in our ablation study. Subsequently, these patches are upscaled back to the original dimensions of $512 \times 512$ with Bicubic Interpolation. Each upscaled patch and the initial image $X$ are then used as an input pair to train a Siamese network using the default cross-entropy loss function augmented with regularization (Koch et al., 2015), with the ground truth similarity set to one when the patch contains any tumor (and thus similar to the original image) and zero otherwise. The trained Siamese network's output, $y_i$, then reflects the likelihood that patch $x_i$ contains any tumor. After this, the attention map of the patch $x_i$, $y_i'$, is calculated, as:

$$y_i' = g(y_i) \cdot c \cdot J, \tag{1}$$

$$g(y_i) = \frac{y_i - \min(Y)}{\max(Y) - \min(Y)}, \tag{2}$$

where $g(y_i)$ is a normalization function, with $y_i$ reflecting the predicted similarity score relative to the original image, $c$ is a constant for attention intensity, $J$ is an $m \times m$ ma-

trix filled exclusively with ones that reconstruct the attention maps to patch size, and $Y = \{y_1, y_2, \ldots, y_n\}$ is the set of predicted similarity scores for all the small patches cropped from the input image $X$. The normalization function $g(y_i)$ is a key, as the Siamese network may miss tiny tumors, causing many of the $y_i$ scores in $Y$ to approach 0. Thus, directly using these unnormalized scores for attention would be quite ineffective. Nevertheless, small tumors typically stand out more than the adjacent areas, resulting in slightly higher $y_i$ scores. Normalizing these scores with $g(y_i)$ to the range $[0, 1]$ enhances attention to the areas with small tumors. The subsequent multiplication by the attention intensity $c$ is for accommodating the input normalization used in YOLO. Afterwards, these maps $y_i'$ are stitched together based on the locations of the corresponding patches in the original image $X$, to form a size $512 \times 512$ attention map, $Y'$, matching the original image's dimensions. Finally, $Y'$ is concatenated alongside the input image $X$ with the matrix concatenation function $\sigma$ as an additional channel, forming the attention-augmented input, $X'$, for YOLOv8x (see Fig. 2(b)).

An alternative design to Eqs. (1) and (2) is, instead of using the normalized scores $y_i$ and the attention intensity constant $c$, to directly combine the Class Activation Map (CAM) (Zhou et al., 2016) from the small patches into a large $512 \times 512$ CAM matrix, and use it as the attention map $Y'$. However, our experiments revealed that this strategy does not significantly enhance the performance of the baseline detection models. The details and implications of this finding will be further explored and elucidated in the ablation study section.

### 3.2. Detection Head:

YOLO (You Only Look Once) (Redmon et al., 2016), standing out as a leading model in the realm of object detection, has undergone significant advancements through various versions. Despite their original design for natural images, YOLO models from YOLOv5 onwards have integrated auto-anchor algorithms, improving their detection of smaller objects beyond the capabilities of many traditional detection models (Jocher, 2020). The efficacy of YOLO in detecting brain and bone tumors has been demonstrated in studies such as RCS-YOLO (Kang et al., 2023) and YOLO-DL (Li et al., 2023). Given the similarities in the challenges posed by liver tumor detection, YOLO is selected as the preferred model for our detection head. Further, we opt for YOLOv8x (Jocher et al., 2023), the most recent version known for its state-of-the-art capabilities, and train it utilizing our attention-augmented input $X'$.

## 4. Experiments

### 4.1. Experimental Setup:

In the attention module, we choose a patch-cropping size $m$ of 64, tailored to the typical tumor sizes observed on our dataset. We adjust the attention intensity constant $c$ to 255, complementing the subsequent image normalization by YOLOv8x. For the Siamese network (Koch et al., 2015), we employ the Adam optimizer (Kingma and Ba, 2014) combined with a Binary Cross Entropy loss function (Ba and Caruana, 2014), setting the batch size to 16 and the learning rate to $1e-4$, with an input image size of $512 \times 512$. To address the

imbalance in the patch dataset fed to the Siamese network, Standard Scale Jittering (Ghiasi et al., 2021) is utilized to augment 100% of the positive patches and 20% of the negative patches. The training is conducted in 200 epochs, with an early stopping mechanism triggered after 50 epochs of no performance improvement. For the YOLOv8x detection head, we enhance the default data augmentation strategies by setting the mixup augmentation rate to 0.1 and the mosaic augmentation rate to 0.3, aiming to prevent overfitting and improve model robustness. The learning rate for YOLOv8x is $5e - 5$, with the training duration extended to 2,000 epochs and an early stopping criterion of 100 epochs. All the remaining settings are maintained with their default values.

### 4.2. Baseline Selection:

For baseline comparisons, we select leading models across various categories: nnU-Net (Isensee et al., 2021) as a superior version of U-Net (Ronneberger et al., 2015), SPN-TS (He et al., 2023) which utilizes an FPN (Lin et al., 2017) architecture for detecting small breast tumors, and Transformer-based models Swin-Unet (Cao et al., 2022) and RT-DETR (Lv et al., 2023). Additionally, RCS-YOLO (Kang et al., 2023), designed for brain tumor detection, and the latest YOLO version, YOLOv8x (Jocher et al., 2023), are also included to showcase the cutting-edge in YOLO advancements. Moreover, we conduct further assessments of our PCA module's effectiveness by substituting it with other state-of-the-art attention mechanisms. These include the self-attention mechanism from ViT-YOLO (Zhang et al., 2021), the multi-attention mechanism from Multi-attention Tri-branch Network (MTNet) (Zhong et al., 2023), and channel-wise attention (Li et al., 2020). Each of them is integrated with the same detection head, YOLOv8x, under identical hyper-parameter configurations. Nevertheless, certain attention methods published recently are incompatible with our dataset, such as the slice-wise attention tailored for 3D datasets (Lu et al., 2023) and the cross-attention designed for multi-modal datasets (Lin et al., 2023), and thus are not included in the comparisons.

All the experiments are run on 3 NVIDIA A100 GPUs with 40GB memory each, and 5-fold cross-validation is performed to ensure generalizability. As this is a detection task, we use the mAP@50% score, Precision, and Recall as the evaluation metrics.

## 5. Results and Discussions

Table 1 presents the mAP@50%, Precision, and Recall scores for all the evaluated models. As one can see from Table 1, our PCA-YOLO outperforms all the other models in all of the mAP@50%, Precision, and Recall metrics, and it also demonstrates the lowest standard deviations across these metrics, which validate its better stability. PCA-YOLO surpasses the top-performing baseline, MTNet (Zhong et al., 2023) combined with YOLOv8x, by an average mAP@50% score of 2.3%, indicating that our proposed PCA module is more effective than other attention mechanisms. A comparative analysis between the original YOLOv8x and its attention-augmented variants reveals that on our dataset, different attention mechanisms improve YOLOv8x's detection capabilities to varying degrees: Channel Attention by 0.2%, Self Attention by 1.1%, MTNet by 1.2%, and our PCA module by 3.5%.

It is also interesting to note that YOLO-based models generally outshine those in the other categories on our dataset, including SPN-TS (He et al., 2023), tailored for small breast

Table 1: mAP@50%, Precision, and Recall scores of different models on the SLTD dataset. Four types of models, including U-Net, Feature Pyramid Network (FPN), Transformers (XFMR), and YOLO, are included for comparison. The last column presents the paired t-test p-value for the mAP@50% results of each baseline model compared to our PCA-YOLO model.

| Type | Model | $mAP_{50}$ (%) | Precision (%) | Recall (%) | p-value |
|------|-------|------------|---------------|------------|---------|
| U-Net | nnU-Net | $36.1 \pm 4.7$ | $56.2 \pm 6.5$ | $33.5 \pm 5.8$ | 0.000 |
| FPN | SPN-TS | $70.8 \pm 4.6$ | $80.9 \pm 6.3$ | $63.8 \pm 7.7$ | 0.018 |
| XFMR | Swin-Unet | $55.8 \pm 6.5$ | $67.5 \pm 10.1$ | $52.0 \pm 10.4$ | 0.002 |
| | RT-DETR | $66.5 \pm 3.6$ | $77.0 \pm 5.1$ | $61.8 \pm 6.2$ | 0.001 |
| YOLO | RCS-YOLO | $72.5 \pm 5.0$ | $74.1 \pm 8.4$ | $64.7 \pm 6.7$ | 0.012 |
| | YOLOv8x | $73.7 \pm 3.4$ | $80.8 \pm 5.5$ | $66.3 \pm 4.9$ | 0.035 |
| | Channel-Attention + YOLOv8x | $73.9 \pm 3.3$ | $80.5 \pm 5.7$ | $68.9 \pm 4.8$ | 0.031 |
| | Self-Attention + YOLOv8x | $74.8 \pm 3.4$ | $79.6 \pm 5.5$ | $69.1 \pm 4.7$ | 0.052 |
| | MTNet + YOLOv8x | $74.9 \pm 2.9$ | $81.9 \pm 5.3$ | $69.2 \pm 4.7$ | 0.064 |
| | PCA-YOLO (ours) | $\mathbf{77.2} \pm 2.1$ | $\mathbf{82.5} \pm 3.0$ | $\mathbf{71.0} \pm 2.4$ | / |

Table 2: Ablation study on the proposed PCA module.

| Model Structure | $mAP_{50}$ (%) | Precision (%) | Recall (%) |
|-----------------|------------|---------------|------------|
| YOLOv8x | $73.7 \pm 3.4$ | $80.8 \pm 5.5$ | $66.3 \pm 4.9$ |
| YOLOv8x w/ CAM attention | $74.5 \pm 3.6$ | $80.6 \pm 4.1$ | $67.7 \pm 4.5$ |
| PCA-YOLO w/o $g(y_i)$ | $75.6 \pm 3.9$ | $81.6 \pm 4.8$ | $70.5 \pm 3.7$ |
| PCA-YOLO w/ $128 \times 128$ patches | $75.9 \pm 2.3$ | $81.6 \pm 3.0$ | $70.8 \pm 2.8$ |
| PCA-YOLO w/ $256 \times 256$ patches | $76.7 \pm 2.5$ | $81.4 \pm 3.6$ | $70.4 \pm 2.1$ |
| PCA-YOLO w/ $64 \times 64$ patches | $\mathbf{77.2} \pm 2.1$ | $\mathbf{82.5} \pm 3.0$ | $\mathbf{71.0} \pm 2.4$ |

tumor detection, and RT-DETR (Lv et al., 2023), which outdoes YOLO in natural image detection. This underscores YOLOv8's adeptness in small liver tumor detection.

Furthermore, Table 1 reveals that all the evaluated models exhibit lower Recall scores relative to Precision, a trend attributable to the dataset's exclusive composition of small-sized tumors, which are inherently more challenging to detect than their normal-sized counterparts. This discrepancy underscores the significant difficulties in identifying small liver tumors. Nevertheless, our model stands out as the only one achieving a Recall score above 70.0%, outperforming the second-best model by 1.8%. This distinction highlights our model's effectiveness in addressing the task of small liver tumor detection compared to the other models. Fig. 3 visualizes PCA-YOLO's successful detection examples. The siamese network's accuracy and more visual comparison results with the other methods are provided in the Appendix A, Supplementary Material.

We also conducted a paired t-test comparing PCA-YOLO against all baseline models, with p-values reported in Table 1, using $\alpha = 0.05$. As shown, our model demonstrates statistically significant improvements over the majority of baselines. These findings indicate that PCA-YOLO consistently outperforms existing methods, with strong statistical evidence supporting its effectiveness.

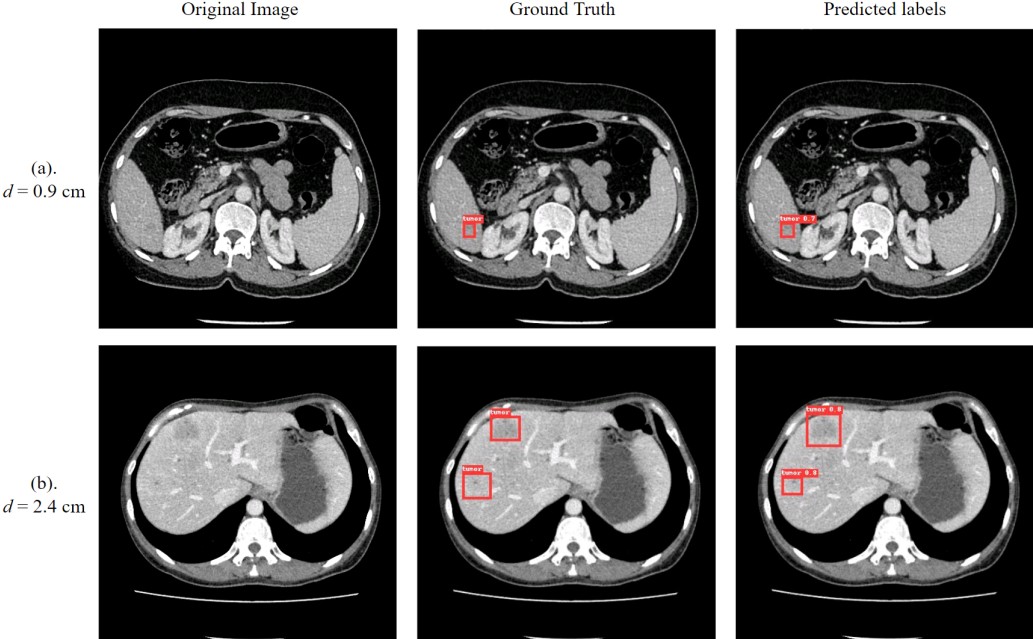

Figure 3: Visualization examples of our detection results with (a) a tumor of bounding box size $d = 0.9$ cm and (b) a tumor of maximum bounding box size $d = 2.4$ cm.

## 5.1. Ablation Study of the PCA Module:

As previously discussed, we explored an alternative approach by directly applying the Class Activation Map (CAM) (Zhou et al., 2016) from the Siamese network as attention maps, rather than using normalized patch attention. However, as indicated by Table 2, this method results in only a 0.8% increase in the mAP@50% score, significantly less than the 3.5% enhancement achieved by PCA-YOLO. This suggests the superiority of our proposed PCA module over CAM-based attention, possibly due to the patch-based attention's compatibility with YOLO's anchor box mechanism. Further, an ablation study on the PCA module's normalization function, $g(y_i)$, highlights its critical role. As shown in Table 2, omitting this normalization step leads to reductions in the mAP@50%, Precision, and Recall scores by 1.6%, 0.9%, and 0.5%, respectively, compared to the original PCA-YOLO model, underscoring the significance of the normalization process in our PCA module. Lastly, an ablation study is conducted to evaluate the impact of varying patch sizes. As demonstrated in Table 2, a patch size of $64 \times 64$ attains superior performance compared to the configurations with the other two patch sizes.

## 5.2. Error Analysis

Examples of detection errors are presented in Fig. 4. In Case (a), a small tumor was incorrectly merged with an adjacent larger tumor, despite medical annotations indicating them as separate entities. This may be due to the small tumor containing too few distinguishable features, making differentiation inherently challenging for the model. Although our model

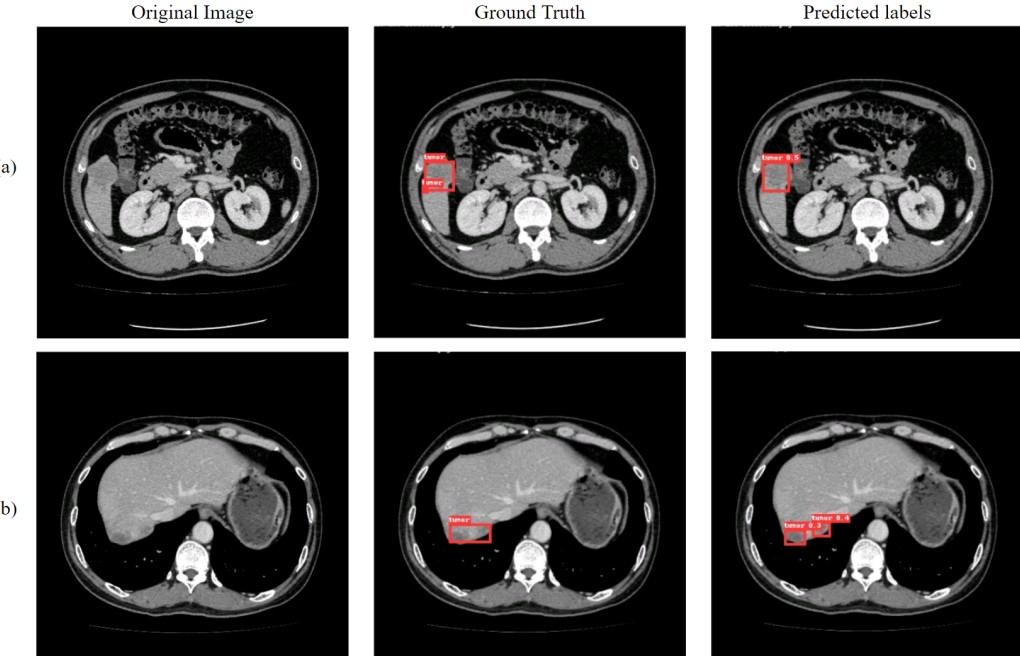

Figure 4: Examples of detection errors.

successfully detects most small tumors, rare cases like this suggest that further refinement in feature extraction could enhance robustness. Conversely, in Case (b), a single tumor was mistakenly detected as two distinct tumors, despite the model correctly capturing its location. This misclassification may be attributed to inconsistencies in texture or intensity variations within the tumor, leading to over-segmentation. While our model demonstrates strong overall detection performance, incorporating improved spatial feature aggregation in future iterations could further mitigate such segmentation inconsistencies.

## 6. Conclusions

In this paper, we proposed PCA-YOLO, a novel detection framework developed upon YOLOv8x, dedicated to addressing the challenges of detecting small-sized liver tumors — a task that, besides being more challenging than identifying liver tumors of normal sizes, holds significant clinical importance. Our new PCA-YOLO model surpasses existing state-of-the-art detection methods in the realm of small liver tumor detection with an mAP@50% score of 77.2%. Furthermore, the PCA module we developed demonstrated superior performance over alternative attention mechanisms when integrated with the same detection architecture. To support this specialized detection task and for future follow-up research from the medical imaging community, we have compiled the SLTD dataset, which consists of 208 3D CT volumes, encompassing 41,587 2D slices, with 452 slices annotated with bounding boxes by our team of skilled radiologists.

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

## Appendix A. Supplementary Material

Table 3: Accuracy of the Siamese network using patch $x_i$ and original image $X$ as inputs with patch size of $64 \times 64$.

| Model | Accuracy(%) |
|---|---|
| Siamese (Koch et al., 2015) | 93.4 ±2.5 |

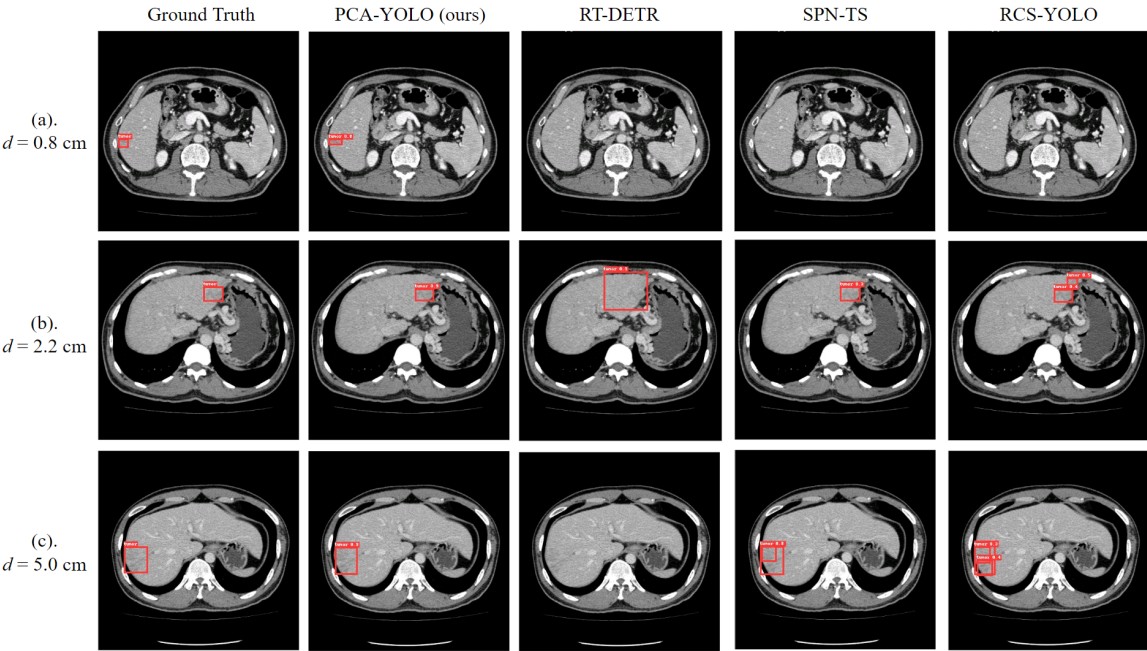

Figure 5: Visual comparisons of PCA-YOLO (ours) with RT-DETR (Lv et al., 2023), SPN-TS (He et al., 2023), and RCS-YOLO (Kang et al., 2023) for: (a) a small-sized tumor with bounding box size $d = 0.8$ cm, (b) a median-sized tumor in our dataset with bounding box size $d = 2.2$ cm, and (c) the largest tumor in our dataset with bounding box size $d = 5.0$ cm.

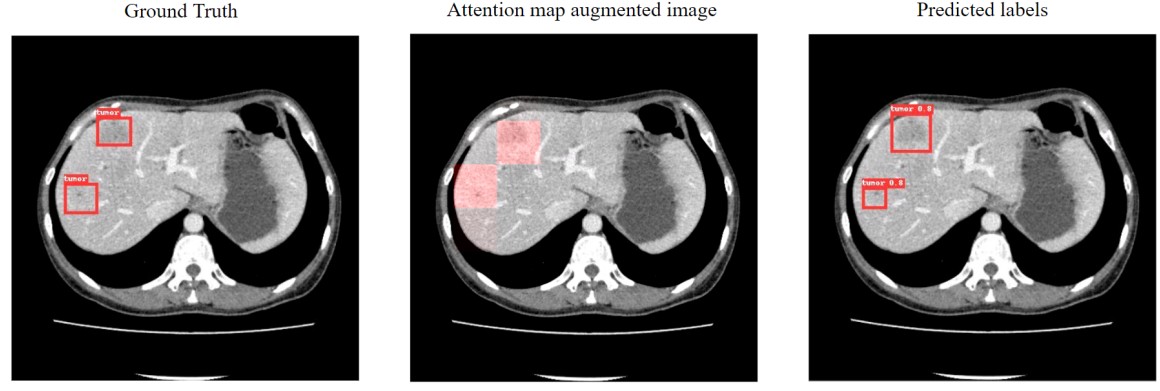

Figure 6: Example of an input image augmented by crop attention map.

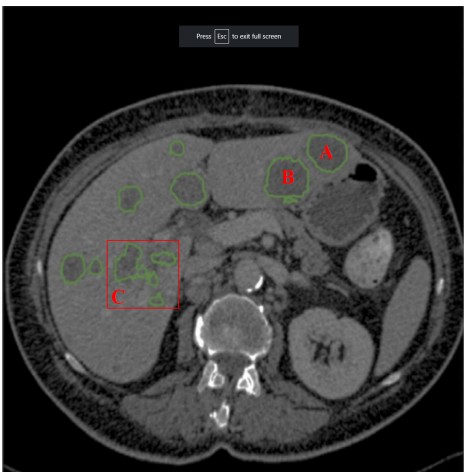

Figure 7: Example of an image from the LiTS dataset. Automatic bounding box extraction from segmentation masks may lead to ambiguities. For example, tumor A and B could potentially be encompassed within a single bounding box while they are different tumors. Moreover, determining whether adjacent tumors in the cluster C should be treated as a single large tumor or as separate tumors requires expert medical judgment. This distinction is crucial, as tumor count is a key parameter in HCC staging according to the RECIST standard and BCLC guidelines.

