# OpenReview forum: "PCA-YOLO: A Small Liver Tumor Detection Model with Patch-Contrastive Attention"
_MIDL.io/2025/Conference — MIDL 2025 Poster_

### Official Review · Reviewer_CyHC · 2025-02-15

**Confidence:** 4
**Preliminary Rating:** 3
**Recommendation:** Poster
**Final Rating:** 4

**Summary:**

This paper introduces PCA-YOLO, a novel detection framework for small liver tumors that combines patch-contrastive attention with YOLOv8x.  The main innovation lies in its PCA module that processes image patches through a Siamese network and applies normalization to generate attention maps before feeding into YOLOv8x. The authors also contribute a specialized Small Liver Tumor Detection (SLTD) dataset. Through comprehensive experiments against state-of-the-art methods and ablation studies, the proposed model achieves superior performance.

**Strengths:**

The strengths are summarized in the 2 aspects.
1. Technical Innovation
The PCA module's design is well-motivated, combining Siamese networks with attention mechanisms. The design is well explained and specifically tailed for the small tumor detection with contrast attention mechanisms.
2. Dataset Contribution
Addresses limitations of existing datasets (like LiTS) for tumor detection, and create and plan to release a specialized dataset (SLTD) focused on small liver tumors. This dataset offers practical value for future research in this domain.

**Weaknesses:**

I can see 2 major weakness
1. Limited Analysis of Model Complexity
There is no discussion of inference time and computational overhead. The upsampling of each path for each detection seems computationally expensive. For each detection, it will run upsampling 8*8 (512/64) times. There is no comparison of runtime with other methods.
2. Dataset Limitations
While the dataset focuses on small tumors (1.0-6.5cm), there's no analysis of performance variation across different tumor sizes. If the dataset can divide tumor size into different ranges, which can serve better to analyze the performance on the tumor sizes.
There is no discussion on the dataset bias as the dataset size is still relative small.

**Detailed Comments:**

The detailed comments:
1. Methodology Clarifications
- Describe what is the advantage of YOLOv8x over other YOLO variants.
- Clarify if overlapping patches were considered as an alternative.
- Siamese network is used but lack of details
2. Experimental Details
- The proposed model complexity analysis, total training time, and inference performance is needed
- Add visualization of attention maps to show how PCA module works
- Add error analysis with examples of failure cases
- How do you split dataset into training, validation and test?

**Justification Of The Final Rating:**

Thanks for the authors to address some of my concerns about YOLO selection, complexity and efficiency analysis, and error analysis. This is critical to understand the model with fair comparision. I believe that is a valuable research work to the small liver tumor detection in the practice.

**Justification Of The Preliminary Rating:**

Justification of the rating:
While there are a few limitations (computational complexity, lack of theoretical analysis), these don't block the paper's core contributions. The work presents a clear advance in small tumor detection, combining with technical innovation and practical clinical value.
1. Clinical Significance
- Small liver tumor detection is a critical challenging medical problem.
- The focus on small tumors (1.0-6.5cm) is particularly valuable as these are harder for radiologists to detect.
- The proposed method achieves state-of-the-art recall.
2. Technical Innovation
- Propose a well-designed solution (PCA-YOLO) with well-explained motivation.
- The architecture choices are well-justified and systematically evaluated.
3. Research Contribution
- Introduce a new dataset (SLTD) specifically for small liver tumor detection.
- Provide comprehensive comparisons across different architectural approaches.

**Questions To Address In The Rebuttal:**

Questions to address in the rebuttal.
1. Model Complexity and Computational Efficiency
- What is the model complexity (e.g., parameters, FLOPs)? How does it compare with other models?
- What is the total training time?
- What is inference time per image? Have it been tested on edge device?
2. Generalization Capability
- The dataset is relative small. There are no details how the dataset is created. e.g., Are the images created from different machines?
- Has the model been tested on external datasets?
- How does it perform on very small tumors (<1.0cm)?

**Special Issue:**

Yes

---

> ### Author Response · Authors · 2025-03-08
> **Response to Reviewer's Feedback**
>
> We appreciate the reviewer’s valuable insights and constructive feedback. Below, we provide detailed responses to each question, addressing concerns and clarifying key aspects of our work.
>
> **Q1. Advantage of YoloV8x**
>
> We selected YOLOv8 because it offers the best balance of accuracy and efficiency among YOLO variants. Compared to previous versions, YOLOv8 incorporates an improved backbone, enhanced feature fusion, and anchor-free detection, leading to superior performance in object detection tasks. These improvements enable it to achieve the highest performance among all YOLO variants on our dataset.
>
> **Q2. Overlapping patches**
>
> Our choice of a non-overlapping cropping strategy is inspired by the Vision Transformer's patch-based processing approach, which effectively captures spatial features without redundancy. We plan to evaluate overlapping cropping techniques in future work.
>
> **Q3. Data split**
>
> We employed 5-fold cross-validation for model evaluation. In each fold, the data was split into 80% for training and 20% for testing. This 5-fold cross-validation strategy ensures that each single case is tested during this process, providing a more reliable and robust estimate of model performance.
>
> **Q4. Complexity and Efficiency**
>
> Our Siamese network has approximately 178.5 GFLOPs and 268M parameters, utilizing 15.7GB of memory on a 40GB A100 GPU. YOLOv8x has 258.2 GFLOPs and 68.2M parameters, requiring 12.7GB on the same GPU. Training efficiency was evaluated using 5-fold cross-validation. Per fold, the Siamese model converges in approximately 2.5 hours, while YOLOv8x requires about 2 hours, resulting in a total runtime of \~22.5 hours for the full 5-fold training process. The additional computational overhead primarily comes from the Siamese network itself, as it processes both the original image and its cropped regions for attention computation. However, despite this added step, the inference time per image remains under 1 second: the Siamese network requires \~900ms for attention generation, while YOLOv8x performs detection in \~10ms. This entire inference process (0.91s) remains significantly faster than manual evaluation by radiologists.
> Our model has not yet been tested on an edge device, but we recognize this as an important direction for future work.
>
> **Q5. Visualization of attention maps**
>
> Added in Appendix Figure 6.
>
> **Q6. Error analysis**
>
> Added in revised manuscript as suggested
>
> **Q7. Siamese network’s detail**
>
> We utilize the structure of the original Siamese network while adapting it for our task by modifying the input encoders to process 512×512 images. The model is trained using the Adam optimizer combined with a Binary Cross Entropy loss function, with a batch size of 16 and a learning rate of 1e-4. We provide labeled patch-image pairs, where patches containing tumors are classified as 1 and non-tumor patches as dissimilar 0.
>
> **Q8. Additional dataset details**
>
> Please refer to the revised manuscript.
>
> **Q9. External dataset**
>
> Our dataset is specifically designed as a benchmark for liver tumor detection, whereas the LiTS dataset was created for segmentation tasks. While bounding boxes can be extracted from segmentation masks, this process is not straightforward. As illustrated by the example in Figure 7 (Appendix) of the revised manuscript, in cases where multiple tumors are in close proximity, such as tumor A and tumor B on the image, automatic bounding box extraction from segmentation masks may lead to ambiguities, as these tumors could be encompassed within a single bounding box. Additionally, determining whether adjacent tumors in the cluster C should be treated as a single large tumor (one bounding box) or as separate tumors (multiple bounding boxes) requires expert medical judgment. This distinction is crucial, as tumor count is a key parameter in HCC staging according to the RECIST standard and BCLC guidelines. This underscores the significance of our expert-annotated SLTD dataset. Moving forward, we plan to have our medical team manually annotate bounding boxes on the slices of the LiTS dataset containing small tumors to enable a fair evaluation of our model, which we consider an important direction for future work.
>
> **Q10. Tumors smaller than 1cm**
>
> We acknowledge the potential for confusion between tumor diameter and bounding box size, as the terminology was updated in the revised manuscript. If the reviewer is referring to the original definition based on tumor diameter (<1.0 cm), we clarify that such cases were not encountered in our dataset. However, as our dataset continues to expand with new clinical cases, we will update it accordingly if such cases arise in the future.
>
> If the reviewer is referring to performance on cases with bounding box size <1.0 cm – which represents the cross-section of the tumor in the 2D slice rather than the full tumor diameter – we direct them to Fig. 3(a) and Fig. 5(a) (Appendix), which illustrate successful detections of such cases.

---

### Official Review · Reviewer_n6xX · 2025-02-17

**Confidence:** 4
**Preliminary Rating:** 3
**Final Rating:** 4

**Summary:**

This paper presents an approach that addresses a clinical problem, namely the detection of small liver tumors. The authors propose a PCA (patch contrastive attention) module coupled to the YOLO framework to detect small tumors in liver CT scans. Ablation studies were carried out to understand the impact of PCA module components and patch size on the model. The results show that the proposed model outperforms some detection methods from the literature on the internal dataset. The authors indicate that the dataset will be made publicly available. The article does not present results on a public dataset, in particular the LiTS dataset, which is highly relevant to the task presented.

**Strengths:**

- Addressing the problem of small tumor detection in liver CT: a highly relevant clinical problem.
- The acquisition of a data set of 208 CT scans, which the authors declare to make publically available , is of great interest to the community in order to advance research in this field.
- Ablation studies of the PCA module provide a better understanding of its impact on model performance.
- The paper is clear and well organized.

**Weaknesses:**

- Missing an evaluation on a public dataset, in particular the LiTS dataset, which is highly relevant to the task presented.
- Lack of details on the dataset used: 208 CT scans, annotated by radiologists (what type of expertise, how many annotators per scan? all data from the same center? what type of acquisition phase? image resolution?)
- The notion of similarity between patches and images is ambiguous in the paper: what does “and thus similar to the original image” mean? An enlarged image patch, whether pathological or not, is “similar” to the original image, since it is part of it. The original image contains both healthy and pathological parenchyma. The notion of similarity is ambiguous.
- Attention to paragraph numbering: 3.0.1, 3.0.2 should be 3.1, 3.2. Same for paragraphs 4 and 5.

**Detailed Comments:**

- An evaluation on the LiTS dataset is needed to better show the impact of the proposed method, especially as it fits well with the proposed task. The authors state that for LiTS “it poses challenges in converting segmentation masks to bounding boxes,
especially in complex scenarios involving tumor clusters or adjacent tumors”, but I believe that a simple extraction of bounding boxes from segmentation masks may be a straightforward solution that will allow the authors to evaluate on a public dataset.
- Making the dataset publicly available is much appreciated by the community. Adding details is also very useful to better understand its characteristics (what type of expertise, how many annotators per scan? all data from the same client? what type of acquisition phase? image resolution?).
- Notion of small tumors: the smallest tumors in the dataset are 10 mm in diameter, which is also the threshold for tumor characterization according to RECIST 1.1 guidelines. However, in a clinical workflow practice, a tumor detected at any given time, even with a diameter of 10 mm, forces radiologists to examine the patient's previous examinations to see if this tumor existed before, and thus to detect tumors with a diameter of less than 10 mm. This kind of tumor can also be seen in public datasets such as the LiTS dataset. An evaluation presenting the model's behavior on this kind of tumor will be further beneficial to show the impact of the proposed method.

**Justification Of The Final Rating:**

I would like to thank the authors for providing more comprehensive details during the rebuttal period. This paper addresses the challenge of detecting small liver tumors in CT scans using a YOLOv8x model. The authors also plan to publicly release the dataset, which I believe will contribute to the community’s efforts in developing algorithms for small tumors detection. While extracting bounding boxes from segmentation masks for evaluation on the public LiTS dataset is not a straightforward task, it can still serve as a baseline for comparison. Incorporating this could further strengthen the paper. Therefore my final rating is Weak accept.

**Justification Of The Preliminary Rating:**

This work involves the use of a patch contrastive attention model and its coupling with a YOLO framework to improve the detection of small tumors in CT scans. The methodology used appears to improve detection results over other detection approaches on the internal dataset. According to authors, this dataset will be made publically available, but further details and documentation on this dataset should be provided. An evaluation on the LiTS public dataset is lacking and could be very useful to better understand the model's performance.

**Questions To Address In The Rebuttal:**

- How does the proposed model works on other datasets, namely the LiTS public dataset? even if by simply extracting bounding boxes from the segmentation masks.
- Can you add more details on the characteristics of the used dataset?
- Can you ealborate more on the notion of similarity between the pathological patch and the whole image? what does it actually reflect?

---

> ### Author Response · Authors · 2025-03-08
> **Response to Reviewer's Feedback**
>
> We appreciate the reviewer’s valuable insights and constructive feedback. Below, we provide detailed responses to each question, addressing concerns and clarifying key aspects of our work.
>
> **Q1. Evaluation on the external dataset**
>
> Our dataset is specifically designed as a benchmark for liver tumor detection, whereas the LiTS dataset was created for segmentation tasks. While bounding boxes can be extracted from segmentation masks, this process is not straightforward. As illustrated by the example in Figure 7 (Appendix) of the revised manuscript, in cases where multiple tumors are in close proximity, such as tumor A and tumor B on the image, automatic bounding box extraction from segmentation masks may lead to ambiguities, as these tumors could be encompassed within a single bounding box. Additionally, determining whether adjacent tumors in the cluster C should be treated as a single large tumor (one bounding box) or as separate tumors (multiple bounding boxes) requires expert medical judgment. This distinction is crucial, as tumor count is a key parameter in HCC staging according to the RECIST standard and BCLC guidelines. This underscores the significance of our expert-annotated SLTD dataset. Moving forward, we plan to have our medical team manually annotate bounding boxes on the slices of the LiTS dataset containing small tumors to enable a fair evaluation of our model, which we consider an important direction for future work.
>
> **Q2. Additional dataset details**
>
> The dataset is a retrospective collection from real-world clinical settings, used for liver cancer diagnosis. It has been annotated by radiologists with at least five years of clinical experience. On average, there are 1.49 annotations per slice, depending on the number of tumors present. The images are obtained from different patients' 3D volumes, with each patient contributing an average of 2.2 slices (452 slices from 208 volumes). The CT scans are acquired during the portal venous phase, as it is more commonly used and clinically effective for tumor diagnosis compared to the arterial and unenhanced plain scan phases, according to our medical team. The images have a resolution of 512×512 and were acquired from the same CT machine, with a window width of 150, window level of 50, radiation dose 120kV, slice thickness of 1 mm, and slice gap of 0.8 mm. The images underwent manual quality control to exclude any scans with noticeable artifacts or blurriness and to verify the completeness of all slices.
>
> The Dataset section has been updated based on the reviewer's suggestion, incorporating the additional details mentioned above.
>
>
> **Q3. Ambiguity of the similarity**
>
> The key idea behind our approach is that the Siamese network is trained in a supervised manner to distinguish tumor-containing patches from non-tumor patches. While it is true that the original image contains both healthy and pathological regions, our training strategy explicitly sets the similarity label to 1 for patches containing tumors and 0 otherwise. This forces the network to learn discriminative features associated with tumor presence, such as differences in texture, contrast, and edge characteristics relative to the surrounding healthy tissue. The phrase 'similar to the original image' refers to the fact that tumor-containing patches retain key visual characteristics from the full image, but through supervised learning, the model is trained to focus on tumor-specific features, rather than treating all patches as equally similar to the whole image.
>
> **Q4. Paragraph numbering**
>
> Fixed in article.

---

### Official Review · Reviewer_ib4x · 2025-02-22

**Confidence:** 4
**Preliminary Rating:** 3
**Recommendation:** Poster
**Final Rating:** 4

**Summary:**

This work addresses the problem of small liver tumor segmentation in CT. Specifically, the authors are working with 2D slices and YOLOv8x. They have selected and annotated 452 slices with tumors from within a set of 208 volumes and plan to publish this "SLTD" dataset. Their methodological contribution is a patch-based attention definition which uses a Siamese network to estimate the likelihood of each non-overlapping 64x64 patch to contain a tumor. This is then normalized to the range 0…255 and concatenated with the original image as input to the YOLOv8x detection head. The evaluation shows higher average standard detection measures than many other (2D) state of the art methods.

**Strengths:**

Publishing the dataset should be a nice contribution in itself (but this has not been done yet, so it leaves a lot of questions open, such as whether it contains just the few annotated slices or the full 3D volumes).  Based on the limitation that the method is 2D only, the choice of competing methods used for evaluation is relatively large and modern.  The results do look good.

**Weaknesses:**

Overall, I find the setup mathematically not very convincing, but it rather appears to be an ad-hoc composition of models and normalization "until it works good".  For instance, siamese networks use weight sharing on both inputs, but they operate on different scales in this case.  So I wonder if an architecture with two encoders for 64x64 and 512x512 images would not have been more appropriate.

The formulation "[the annotated] slices were chosen based on their proximity to the initial and final slices where a tumor is visible within each 3D volume" does not really define *how* the slices were selected. (First and last slice? Center slice? First and last where the tumor is "visible enough" according to subjective assessment?)

It also is a considerable limitation that the method is 2D only and could not be compared with 3D methods (likely because the whole evaluation is based on 2D annotations only).

The results are presented without statistical significance tests, so the readers must believe that the averages are representative of the overall performance.  Given that the authors have also compared different versions of their model (different patch sizes, with/without normalization), one may expect some bias in the results and take them with a grain of salt.

**Detailed Comments:**

There must be a better way to use the results of the first attention network than to concatenate them with the input images.  Of course, this simplifies the integration and facilitates comparison with many other methods, but it does not feel optimal yet.

It would be interesting to see some of the attention maps in the appendix as well.

Did you also consider overlapping patches?  If not, why not?

*How* did you "upscale" the 64x64 patches to 512x512 for training the Siamese network?  (Missing details in paper)

You could explicitly state that the purpose of the matrix J is for bringing the attention maps back to the original image size.

**Justification Of The Final Rating:**

I appreciate the author's rebuttal which addresses many points.  I also just saw the new appendix figure with a LiTS example addressing the tumor counting argument.  While I still have a number of small questions marks when looking at the paper and rebuttals and in my personal list of reviewed papers and still consider this contribution to be less scientifically clean than others with a rating of 4, I think that the new revision and its introduced statistical testing probably justifies raising the rating to 4 (it's a 3.5 from my POV now).

**Justification Of The Preliminary Rating:**

I think the results look good, and the topic is highly relevant for MIDL, but the mathematical rigor behind this work does not make it as reliable as one would like it to be.  Maybe adding significance tests (in particular in light of the relatively small dataset) could make my rating more positive.

**Questions To Address In The Rebuttal:**

(See weaknesses above, in particular the significance testing.)

In your introduction, you correctly state that "the maximum tumor diameter is a pivotal factor influencing patient prognosis" and then conclude that "detection models stand out as more suitable alternatives … in comparison to segmentation models", which surprised me, because the latter should be much more suitable for providing a RECIST diameter, no?

---

> ### Author Response · Authors · 2025-03-08
> **Response to Reviewer's Feedback**
>
> We appreciate the reviewer’s valuable insights and constructive feedback. Below, we provide detailed responses to each question, addressing concerns and clarifying key aspects of our work.
>
> **Q1. Significance testing:**
>
> In response to the reviewer's suggestion, we conducted a paired t-test comparing PCA-YOLO against all baseline models, with p-values reported in Table 1 of the revised manuscript, using α = 0.05. As shown, our model demonstrates statistically significant improvements over the majority of baselines. These findings indicate that PCA-YOLO consistently outperforms existing methods, with strong statistical evidence supporting its effectiveness.
>
> **Q2. Detection v.s. Segmentation:**
>
> We appreciate the reviewer’s insightful comment regarding the suitability of detection models versus segmentation models for our task. While segmentation models can theoretically provide a more precise delineation of tumor boundaries, we argue that detection models offer distinct practical advantages in clinical applications, particularly for small liver tumor detection and staging. Our rationale is based on the following three key considerations:
>
> 1). Detection models inherently provide the number of tumors through their bounding box outputs, a critical parameter for HCC staging. In contrast, segmentation models require additional post-processing steps to count tumors from predicted masks, which can be challenging when multiple tumors are in close proximity or exhibit connected masks. This added complexity increases the likelihood of segmentation inconsistencies, particularly in cases where adjacent tumors are difficult to separate clearly.
>
> 2). While segmentation models provide precise tumor contours, clinical decisions do not solely rely on segmentation outputs. In practice, radiologists obtain an approximate tumor diameter directly from the bounding box dimensions (width or height) predicted by detection models. Given that both detection and segmentation models are prone to errors—such as partial tumor omission in segmentation outputs—radiologists must manually verify and refine model predictions regardless of the approach. Consequently, the segmentation model’s ability to define exact boundaries may not provide a decisive advantage over detection models for clinical decision-making.
>
> 3). Creating a high-quality segmentation dataset requires significantly more annotation effort compared to detection datasets, as it necessitates detailed manual contouring of each tumor. In contrast, annotating bounding boxes for detection models is substantially less labor-intensive. Based on estimates from our medical team, the time required to annotate a single image for segmentation is approximately 10 times that of detection. Given the increasing reliance on large-scale datasets for deep learning applications, this considerable difference in annotation efficiency makes detection models a more practical choice for real-world deployment, particularly in resource-limited settings where expert annotation time is a limiting factor.
>
> **Q3. Slice Selection:**
>
> As updated in the revised manuscript, the slices were selected by our medical team based on subjective assessment, favoring those near the first and last slices where the tumor remains 'visible enough.' These slices are particularly challenging for radiologists, as tumors tend to be smaller and harder to detect, making them crucial for evaluation. To ensure annotation reliability, only slices with bounding boxes size of at least 0.4 cm were included. Additionally, we do not include cases with bounding boxes larger than 5.0 cm, as they are generally considered easier to diagnose and provide limited computational value for our task.
>
> **Q4. Better encoder and better integration of attention:**
>
> We appreciate the reviewer’s insight. Refining the Siamese architecture and enhancing the integration of attention network outputs are important directions for our future work. Additionally, we hope our dataset serves as a benchmark to facilitate further advancements in this area.
>
> **Q5. Visualization of attention map:**
>
> Added in Appendix Figure 6.
>
> **Q6. Overlapping patches:**
>
> Our choice of a non-overlapping cropping strategy is inspired by the Vision Transformer's patch-based processing approach, which effectively captures spatial features without redundancy. We plan to evaluate overlapping cropping techniques in future work.
>
> **Q7. Upscale details.**
>
> Added details in paper.
>
> **Q8. Purpose of the matrix J**
>
> Revised as suggested.

---

> > ### Comment · Reviewer_ib4x · 2025-03-14
> >
> > With respect to the detection-vs-segmentation model discussion, I personally find the arguments a little contrived, for instance 1) that counting connected components is an extra step does not concern me at all, since it's a well-established tool in any toolbox, and I wonder if it makes sense to consider the number of tumors "a critical parameter for HCC staging" in the sense that counting two confluent lesions as one would change the stating – if that's the case, I would question the medical practice. (But I agree that detection models may be able to count touching lesions better than a connected components algorithm from a binary segmentation mask.)
> > The argument 2) is even more strange: I agree that clinical practice does not measure tumors by segmenting them (which I think is mostly due to effort and feasibility which only recently improved – and that may also affect guidelines eventually), but I disagree that they are measured from bounding boxes – in my experience, maximal RECIST diameters (with or without orthogonal ones) are used, which correspond to some kind of box, but neither necessarily *bounding* boxes, nor *axis-aligned* ones. The imaging coordinate grid's orientation should not have an affect on clinical decisions in my opinion.

---

> > > ### Author Response · Authors · 2025-03-15
> > > **Response to Reviewer's Comment**
> > >
> > > We appreciate the reviewer’s follow-up and clarification on this point. Tumor count is indeed a critical factor in HCC staging, as reflected in widely adopted guidelines such as BCLC, Milan, and UCSF criteria (*Reig et al., 2022; Mazzaferro et al., 1996; Yao et al., 2001*). However, our paper does not aim to debate the relative importance of tumor count in staging. That said, even if tumor count were less critical, we still observe that existing segmentation-based tools struggle to reliably separate adjacent tumors, particularly in cases where tumors are in close proximity. Our concern is not merely about counting errors (i.e., overcounting or undercounting tumors), but rather that segmentation-based tools can mistakenly merge distinct adjacent tumors into a single larger mass, potentially altering the perceived tumor burden.
> > >
> > > We agree that RECIST diameters are used in clinical practice and that they do not necessarily correspond to axis-aligned bounding boxes. However, both 2D segmentation and detection models inherently estimate tumor size within a single slice and do not directly capture the maximal 3D RECIST diameter. This limitation applies regardless of whether bounding boxes or segmentation masks are used. Given that clinical practice does not rely on segmentation masks for direct tumor measurement, the advantage of precise boundary delineation in segmentation becomes less critical. In contrast, bounding boxes provide a reasonable approximation of tumor size within a given slice, offering a practical and efficient way to localize and measure tumors at the 2D level. Addressing the full 3D tumor extent would require additional volumetric reconstruction techniques, which are outside the scope of this study but remain an important direction for future research.
> > >
> > > We sincerely appreciate the reviewer’s insights and thoughtful discussion and hope this clarification adequately addresses the concerns.
> > >
> > >
> > >
> > > [References]
> > >
> > > [1] Maria Reig, Alejandro Forner, Jordi Rimola, Joana Ferrer-Fabrega, Marta Burrel, Angeles ´ Garcia-Criado, Robin K Kelley, Peter R Galle, Vincenzo Mazzaferro, Riad Salem, et al. BCLC strategy for prognosis prediction and treatment recommendation: The 2022 update. *Journal of Hepatology*, 76(3):681–693, 2022.
> > >
> > > [2] Vincenzo Mazzaferro, Enrico Regalia, Roberto Doci, Salvatore Andreola, Andrea Pulvirenti,
> > > Federico Bozzetti, Fabrizio Montalto, Mario Ammatuna, Alberto Morabito, and Leandro
> > > Gennari. Liver transplantation for the treatment of small hepatocellular carcinomas in
> > > patients with cirrhosis. *New England Journal of Medicine*, 334(11):693–700, 1996.
> > >
> > > [3] Francis Y Yao, Linda Ferrell, Nathan M Bass, Jessica J Watson, Peter Bacchetti, Alan
> > > Venook, Nancy L Ascher, and John P Roberts. Liver transplantation for hepatocellu-
> > > lar carcinoma: expansion of the tumor size limits does not adversely impact survival.
> > > *Hepatology*, 33(6):1394–1403, 2001.

---

### Author Rebuttal · Authors · 2025-03-08

**Rebuttal:**

We sincerely appreciate the valuable feedback from the reviewers, as well as the efforts of the program chairs and area chairs in facilitating this review process. Based on the insightful comments provided, we have carefully revised the manuscript to address the concerns raised. All changes are highlighted in the revised manuscript for clarity.

**Supporting Material:**

/attachment/b797587f4d58a7a02a64a4f553edd3a62b62584a.pdf

---

### Meta-Review · Area_Chair_SegY · 2025-03-16

**Recommendation:** Accept (Poster)
**Confidence:** 4

**Metareview:**

This paper presents PCA-YOLO, a novel small liver tumor detection model incorporating Patch-Contrastive Attention, along with the introduction of the SLTD dataset. Reviewers acknowledge its clinical significance, technical novelty, and comprehensive experiments. While initial concerns were raised regarding mathematical rigor, dataset limitations, and computational complexity, the rebuttal effectively addressed these issues through statistical validation, detailed efficiency analysis, and methodological clarifications. Therefore, I recommend accepting this paper.